# The geography of COVID-19 spread in Italy and implications for the relaxation of confinement measures

Enrico Bertuzzo [1,2], Lorenzo Mari [3], Damiano Pasetto [1], Stefano Miccoli [4], Renato Casagrandi [3], Marino Gatto [3] & Andrea Rinaldo [5,6✉]

The pressing need to restart socioeconomic activities locked-down to control the spread of SARS-CoV-2 in Italy must be coupled with effective methodologies to selectively relax containment measures. Here we employ a spatially explicit model, properly attentive to the role of inapparent infections, capable of: estimating the expected unfolding of the outbreak under continuous lockdown (baseline trajectory); assessing deviations from the baseline, should lockdown relaxations result in increased disease transmission; calculating the isolation effort required to prevent a resurgence of the outbreak. A 40% increase in effective transmission would yield a rebound of infections. A control effort capable of isolating daily ~5.5% of the exposed and highly infectious individuals proves necessary to maintain the epidemic curve onto the decreasing baseline trajectory. We finally provide an ex-post assessment based on the epidemiological data that became available after the initial analysis and estimate the actual disease transmission that occurred after weakening the lockdown.

[1] Dipartimento di Scienze Ambientali, Informatica e Statistica, Universitá Ca' Foscari Venezia, 30172 Venezia-Mestre, IT, Italy. [2] Science of Complexity Research Unit, European Centre for Living Technology, 30123 Venice, IT, Italy. [3] Dipartimento di Elettronica, Informazione e Bioingegneria, Politecnico di Milano, 20133 Milano, IT, Italy. [4] Dipartimento di Meccanica, Politecnico di Milano, 20156 Milano, IT, Italy. [5] Laboratory of Ecohydrology, École Polytechnique Fédérale de Lausanne, 1015 Lausanne, CH, Switzerland. [6] Dipartimento ICEA, Universitá di Padova, 35131 Padova, IT, Italy. ✉email: andrea.rinaldo@epfl.ch

Althought the pandemic caused by SARS-CoV-2 is still ravaging most countries of the world[1,2] and containment measures are implemented worldwide[3], a debate is emerging on whether these measures might be partially alleviated, and in case how and when[4–9]. This discussion requires appropriate models that guide decision-makers through alternative actions via scenarios of the related trajectories of the epidemic.

The setup of country-wide epidemiological models[10,11] is particularly challenging for SARS-CoV-2 owing to inapparent infections[12–14], and to the marked spatial heterogeneity of the epidemic spread[11]. For example, in Italy, where the (largely underestimated) reported infections and deaths were, respectively, 207,428 and 28,236 as of May 1, the latitudinal characters of the spread of infections showed marked delays in the beginning of the local outbreaks[11].

To make things even more complicated, empirical evidence suggests that mildly symptoma infectious individuals could be as contagious as symptomatic ones[12,15]. Pre-symptomatic infectious cases are also an important vehicle of infection, as epitomized by the value of the pre-symptomatic transmission parameter, which proves larger than the transmission rates of symptomatic and asymptomatic infections[11]. This is supported by field epidemiological evidence[16–18] and virological findings reporting cases of COVID-19 fueled by strong pre- or oligo-symptomatic transmission[19–21] and shedding[22].

We base our analysis on a recently published, spatially explicit model of the COVID-19 spread in Italy, inclusive of mobility among communities, the timing of infection seeding, mobility restrictions and social distancing[11]. We assume that, for the time being and in the near-term, no imported infections occur from outside the national boundaries. The model is a spatial system of coupled ordinary differential equations that solves in time, and for each of the 107 Italian provinces, the balance of, and the coupled fluxes among, several epidemiological compartments in which the total population of a community is subdivided. Specifically, we describe the dynamics of individuals who are susceptible, latently infected, at peak infectivity, asymptomatic/mildly symptomatic, infected with heavy symptoms, and recovered (see "Methods" section). Local communities are connected by mobility fluxes of individuals from the mobile epidemiological compartments (susceptible, exposed, peak infectivity, asymptomatic/mildly symptomatic, and recovered individuals). Thus, the force of infection of each community (see "Methods" section) depends not only on the local epidemiological variables, but also on those of the connected communities. Infections, therefore, not only do occur within each community, but can also be imported from, or exported to, linked communities. In addition, the model accounts for infections occurring because individuals of different communities meet in a third location. The relative balance of the fluxes among the various compartments is regulated by process parameters that are estimated in a Bayesian framework (see "Methods" section).

The fundamental improvements of our framework with respect to other non-spatial, well-mixed models initially devised for single megacities[23], or for a whole country[24], lie in the detailed description of the geographic context and its networks of epidemiological interactions. Therefore, we have updated the benchmark model[11] through the estimation of parameters using the number of daily hospitalized cases in all 107 Italian provinces from February 24 to May 1 (see "Methods" section). To estimate parameters, we account for the set of progressively more restrictive measures that were introduced from February 22 (initial restrictive measures) to March 22, when Italy went into a full lockdown closing also non-essential industrial and other production activities[25–27].

Available epidemiological data[28–30] must be viewed as an approximation. In fact, confirmed infections depend on testing efforts that local officials were able to deploy to identify confirmed infections, thus leading to under-reporting. The ratio of confirmed to actual infections was estimated to be around 10%[11]. Under-reporting applies even to fatality counts, although to a lesser extent with respect to reported infections[31,32]. Moreover, fatality rate can change in time due to stress in health care facilities[32]. In order to alleviate these problems, in this work we used for parameter estimation only reconstructed data on daily rates of hospitalization (see "Methods" section).

Health-policy and science underpin the design of suitable containment strategies, which include individual and collective (local and medium- to long-distance) mobility limitations[25], provision of personal protective equipment (PPE)[33], massive, possibly targeted identification of infectious cases[34,35], and the setup of layers of administrative and environmental engineering controls[33]. These strategies must consider the level of connectivity realized among communities after lockdown release, and the different epidemiological parameters that effectively characterize them[10,11]. Recent results on the effects of lifting restrictions in the Boston area suggest that a response system based on enhanced contact tracing and testing can have a major role in relaxing social distancing interventions in the absence of herd immunity against SARS-CoV-2[36].

Here, we generate scenarios of the Italian infection dynamics resulting from the bulk effect of lifting the current restrictions, which initiated on May 4. How will the modes of relaxation of previous confinement measures affect residual epidemic trajectories? The answer to this question is not trivial, because different activities will be allowed to resume at different times. In addition, acquired awareness may have different lasting effects on social behavior regardless of imposed measures, and compliance to proper use of PPE[33] may fade away in time. Here, we propose to assess the actual change in overall transmission by tracking the departure of the epidemic curve from the one projected by using the transmission rate achieved during the lockdown. We then address the mitigation of the likely increased exposure, in particular by estimating the sufficient number of case isolation interventions that would prevent rebounding of the epidemics. Finally, we provide an ex-post assessment of the explored scenarios comparing them with the actual space-time progression of the outbreak as measured by the epidemiological data that became available after the initial submission of this study.

## Results

**Parameter estimation and model results**. The model reproduces well the prevalence of cumulative hospitalizations in the 107 Italian provinces up to May 1 (Figs. 1 and 2). By considering heterogeneous transmission rates after March 22 (see "Methods" section), we estimate a large reduction in the effective disease transmission rate in each province. This reduction, expressed as a ratio of effective transmission estimated on May 1 to the initial uncontrolled one, ranges between 0.3 and 0.4 depending on location (Fig. 1d). Technically, this reduction is computed via the product of the reduction in transmission rates ($\beta_{P_3}/\beta_{P_0}$, see "Methods" section) times the fraction of the population still susceptible to the infection on May 1. The latter, however, is very sensitive to the fraction of infections that develop heavy symptoms (parameter $\sigma$ in the model, see "Methods" section). The reference value assumed is $\sigma = 25\%$, which is consistent with empirical evidence[10]. However, we carried out a sensitivity analysis to investigate the role of inapparent infections[12] by repeating parameter estimation with $\sigma = 10\%$ and 50% as well, thus covering a broad enough spectrum of possible values. All other

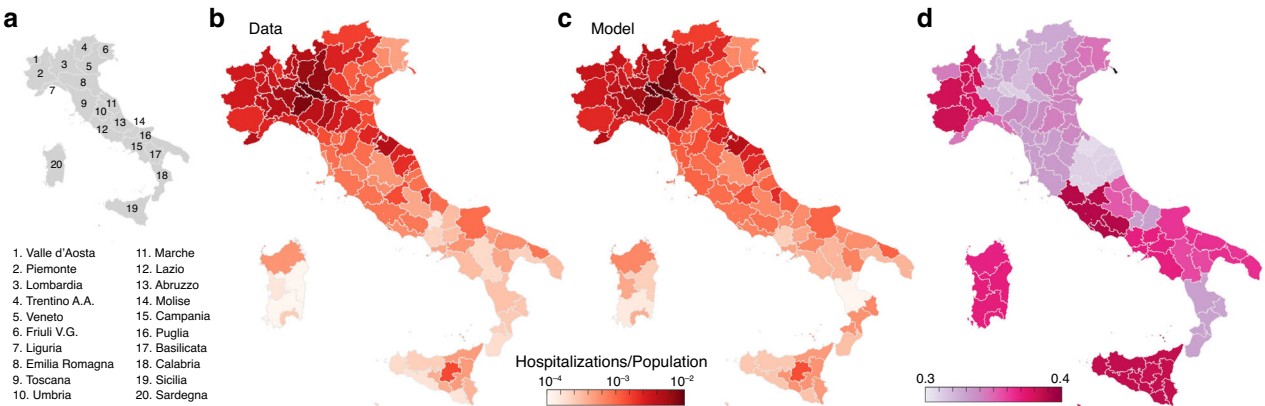

**Fig. 1 Geography of COVID-19 spread in Italy.** The comparative analysis of data and model results for hospitalizations in 107 Italian provinces as of May 1, 2020 is supported by: **a** a sketch of the Italian regions; **b**, **c** the prevalence of cumulative hospitalizations in each Italian province up to May 1, reconstructed data (**b**) and model simulations (**c**); **d** ratio between the estimated transmission rate on May 1, and the one estimated at the beginning of the outbreak (February 24).

**Fig. 2 Daily numbers of newly hospitalized cases for Italy and its hardest hit regions.** Shown here are reconstructed data (empty circles), and model results (solid lines and confidence intervals). Clockwise from top: Italy, Lombardia, Piemonte, Marche, Veneto, and Emilia Romagna. The remaining regions are shown in Supplementary Fig. 1. The blue solid line represents the baseline scenario, i.e., the median of the computed results with transmission estimated during lockdown maintained indefinitely beyond May 3, 2020. The green and purple solid lines represent the scenarios corresponding to a release of the containment measures determining an effective increase in the overall transmission rates of, respectively, 20% and 40%. The 95% confidence intervals are color-coded in analogy to their median scenarios. Plots refer to a fraction of infections leading to heavy symptoms $\sigma$ equal to 25%. Plots referring to the other two values considered ($\sigma = 50\%$ and 10%) are reported in Supplementary Figs. 2 and 3.

**Table 1 Model parameters.**

| Parameter | Units | Median | 95% CI | Information |
|---|---|---|---|---|
| $\beta_{P_0}$ | (d$^{-1}$) | 1.26 | [1.24, 1.28] | Estimated |
| $1/\delta_E$ | (d) | 4.6 | | 15,26 |
| $1/\delta_P$ | (d) | 2 | | 15,26 |
| $1/\eta$ | (d) | 5 | | 52 |
| $1/\gamma_I$ | (d) | 14 | | 11 |
| $1/\alpha_I$ | (d) | 25 | | 11 |
| $\beta_A/\beta_P$ | (1) | 0.022 | [0.020, 0.030] | Estimated |
| $\beta_I/\beta_A$ | (1) | 1 | | 11,12 |
| $\beta_{P_1}/\beta_{P_0}$ | (1) | 0.89 | [0.87, 0.92] | Estimated |
| $\beta_{P_2}/\beta_{P_1}$ | (1) | 0.72 | [0.70, 0.73] | Estimated |
| Mean $\beta_{P_3}/\beta_{P_2}$ | (1) | 0.50 | [0.48, 0.51] | Estimated |
| Standard deviation $\beta_{P_3}/\beta_{P_2}$ | (1) | 0.038 | [0.025, 0.053] | Estimated |
| $\Delta t_O$ | (d) | 35 | | 11 |
| $\omega$ | 1 | 2.42 | [2.33, 2.52] | Estimated |

The posterior distribution of the parameters marked as estimated was sampled through the DREAM$_{ZS}$ implementation of the Markov chain Monte Carlo algorithm[53]. For all estimated parameters, we used uninformative priors within biologically meaningful boundaries. Following our previous application[11], we assumed $\sigma = 0.25$, $r_S = 0.5$, $\zeta = 0.45$, and $r_E = r_P = r_A = r_R = r_S$, whereas $r_I = r_Q = r_H = 0$. Moreover, $\gamma_Q = \gamma_I = \gamma_H$, $\gamma_A = 2\gamma_I$, and $\alpha_H = \alpha_I$.

parameters, whose meaning is detailed in the "Methods" section, are reported in Table 1.

**Scenarios of national and regional epidemic trajectories**. If the transmission rates estimated at the end of the lockdown persisted indefinitely, the epidemic curve would continue to decrease in all Italian regions (baseline scenario, blue curve in Fig. 2), although at different rates. We report for convenience daily hospitalization counts aggregated for administrative regions, although the model accounts for a finer spatial granularity (107 provinces and metropolitan areas, see Fig. 1).

The lockdown in Italy has been relaxed on May 4. Here, we propose to assess the actual increase in overall transmission of the infection by tracking the departure of the residual epidemic curve from the baseline scenario. This allows us to estimate the overall effect of the new exposure caused by the local lockdown relaxations. An increase of 20% in the transmission rate, subsuming the effective combination of economic activities' resumption and modified contact rates, yields a decline milder than that of the baseline for the new daily hospitalization cases in most Italian regions. A 40% increase would instead determine a significant rebound of the epidemic in most regions (Fig. 2, see also Supplementary Fig. 1).

The trajectories shown in Fig. 2 prove robust with respect to the assumed value of $\sigma$, at least for the relatively short projection horizons considered here, which are relevant to contingency planning. Indeed, the curves in Fig. 2, obtained with $\sigma = 25\%$, compare well with those reported in Supplementary Figs. 2 and 3, obtained by assuming $\sigma = 50\%$ and 10%, respectively.

The fraction of susceptible individuals obtained for different values of $\sigma$, the heavy symptomatic fraction, strongly varies throughout the Italian regions (Fig. 3). Since the beginning of the epidemic up to May 1, the susceptible fraction of the population has decreased more markedly in the northern regions, which have been more severely hit by the outbreak, with the minimum values reached in Lombardia (0.97, 0.95, and 0.87 with $\sigma = 50\%$, 25%, and 10%, respectively). By contrast, central and southern regions had minimal reductions of their susceptible fraction. These results bear obvious implications on possible revamping of the epidemics reaching new peaks of dangerous proportions, as it denies any short- or medium-term possibilities to attain herd immunity.

Different assumptions for $\sigma$ result in different values of the infection fatality rate (IFR), defined as the ratio between the

official death count (at a certain date) and the corresponding total number of infections estimated by the model. As of May 1, we estimate an IFR of 4%, 2%, and 0.8%, respectively for $\sigma = 50\%$, 25%, and 10%.

**Isolation effort**. Isolation of cases to counterbalance the possible increase in transmission following the relaxation of the restrictive measures is a conceivable strategy, alternative to extending lockdown or to stop-and-go enforcement of containment measures[6]. The isolation effort critically depends on tracing and testing. Evidence of peak of viral shedding before and right after symptom onset[15] (see also "Methods" section) suggests that isolation is more effective if targeted at incubating individuals, i.e., those in the exposed, $E$, and peak of infectivity, $P$, compartments of the model (according to the parameters reported in Table 1, around 86% of the infections occur through contact with an individual at peak infectivity). We therefore focus on the $E$ and $P$ compartments and estimate the percentage and the corresponding number of individuals that should be isolated daily (Fig. 4, see "Methods" section) to counterbalance a possible increase in transmission resulting from the loosening of the containment measures, thus maintaining the epidemic curve in the decreasing trajectory achieved during the lockdown (blue lines in Fig. 2). In analogy to the analyses presented above, we show results under the three different assumptions about the heavy symptomatic fraction: $\sigma = 50\%$, 25%, and 10%.

Figure 4 also reports the estimated abundances of exposed, $E$, and individuals at peak infectivity, $P$, in the considered regions at the date of the announced new measures (May 4), along with the expected number of new daily symptomatic cases ($C$) predicted by the model. As an example, in Lombardia an increase in transmission of 40% would lead to a rebound of the epidemic curve (Fig. 2). However, for the reference value of $\sigma = 25\%$, daily isolation of about ~1200 out of ~22,000 (~5.5%) individuals belonging to the $E$ and $P$ compartments would effectively counterbalance the increase in transmission and bring back the curve to the baseline scenario (blue curve in Fig. 2). The reported isolation target in terms of the number of individuals to be isolated (left axis of Fig. 4) refers to the necessary effort right after the relaxation of the containment measures. If the epidemic is successfully controlled, and the cases continue to decline (i.e., they follow the baseline scenario), the isolation effort proportionally decreases over time. The isolation effort in terms of

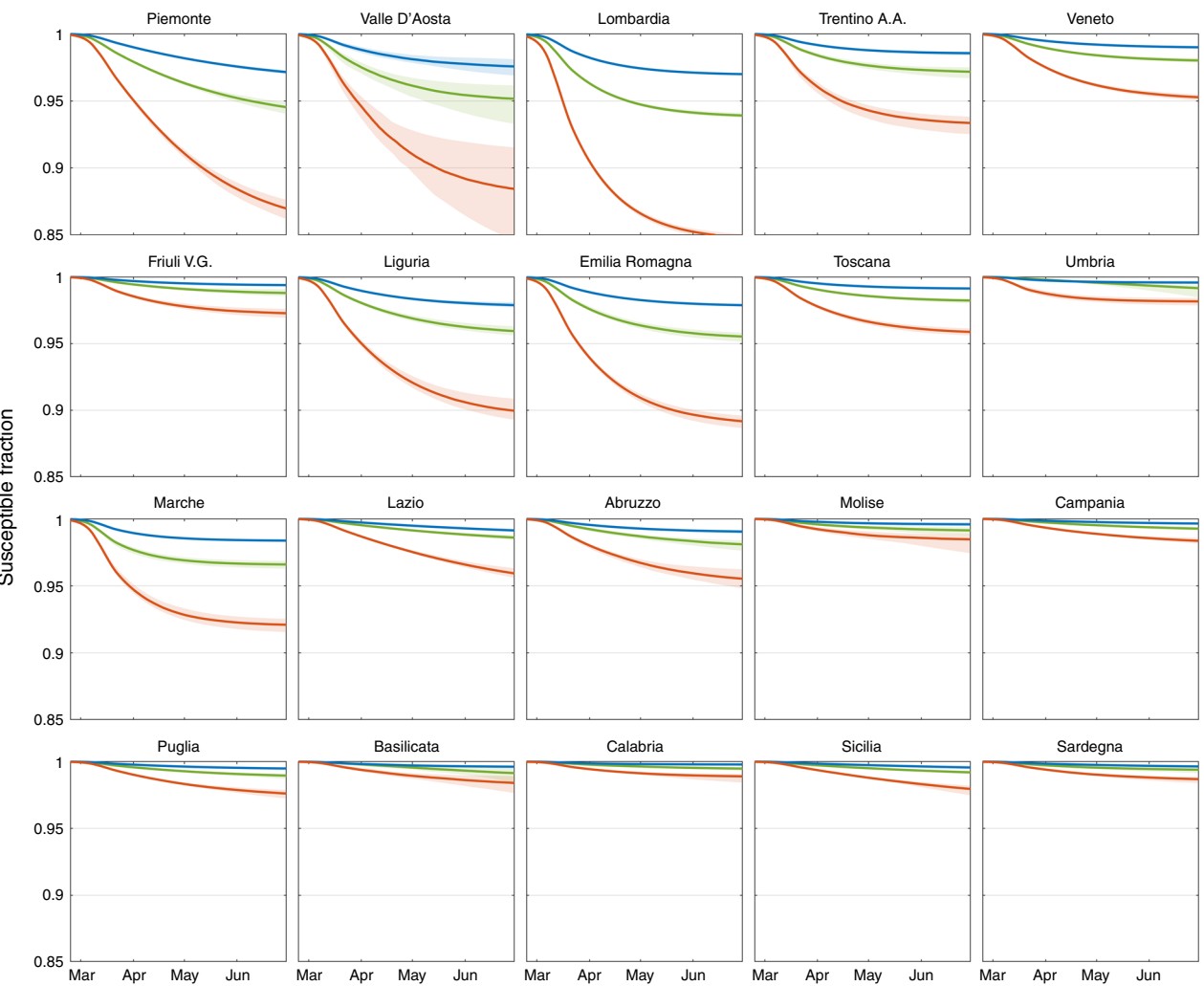

**Fig. 3 Mapping possible acquired immunity in Italy.** Temporal dynamics of the fraction of susceptible individuals in each region, estimated by considering three possible percentages of heavy symptomatic infections: $\sigma = $ 50% (blue curve), 25% (green curve), and 10% (red curve). Thick solid curves refer to medians values, whereas shaded areas indicate the 95% confidence intervals.

percentage of $E$ and $P$ individuals to be isolated (right axis of Fig. 4) remains instead constant.

Although the isolation effort expressed as percentage of the $E$ and $P$ individuals to be isolated daily is not particularly sensitive to the assumed fraction of infections developing heavy symptoms, $\sigma$, the value changes markedly when expressed in terms of the absolute number of individuals (Fig. 4). As $\sigma$ decreases, a larger fraction of the epidemic remains unobserved. Therefore, to closely match the daily hospitalizations data, a much larger pool of $E$ and $P$ individuals is estimated (Fig. 4).

To assess the feasibility of the isolation effort required to contain the epidemic, we report the amount that can be achieved by tracing and isolating all the infections generated by the new daily symptomatic cases (black dashed lines in Fig. 4). For a given increase in transmission, a required effort (solid lines in Fig. 4) exceeding such amount implies that tracing and isolation of all primary infections generated by the new daily symptomatic cases is insufficient. In this case, secondary infections (i.e., infections generated by the primary infectees) need also to be targeted. As the role of the unobserved epidemic increases (i.e., $\sigma$ decreases, from left to right columns in Fig. 4), isolation of primary contacts alone can compensate only for mild increases in transmission.

The timing of the relaxation of the restrictive measures also has a great impact on the isolation effort required to control the

epidemic. Delaying the release of containment measures by an additional month would have reduced the abundance of $E$ and $P$ individuals by about two-thirds, thus proportionally reducing the number of individuals that need be isolated (Supplementary Fig. 7).

**Ex-post assessment**. The analysis presented above was based on data up to May 1, 2020 (see also ref. [37]). Data that became available afterward allows for ex-post assessments of the trajectories projected in Fig. 2. Among the three scenarios explored, the actual progression of the outbreak in the Italian territory after the lifting of lockdown measures is consistent with the baseline scenario (Fig. 5). Some regions, notably Piemonte, exhibit a case count significantly lower than the one predicted by the baseline scenario. To properly quantify such variations in the outbreak dynamics, we re-estimated model parameters including the newly data available (up to June 17, 2020) and an additional parameter controlling the transmission after lockdown was relaxed (see "Methods" section). As the newly available data starts before the end of the lockdown (May 1 vs May 4) and there is an intrinsic delay before variations in transmission appear in hospitalization data, the new parameter estimation exercise enables also an updated evaluation of transmission occurring in the last phase of the lockdown (parameter $\beta_{P_3}$, see "Methods" section).

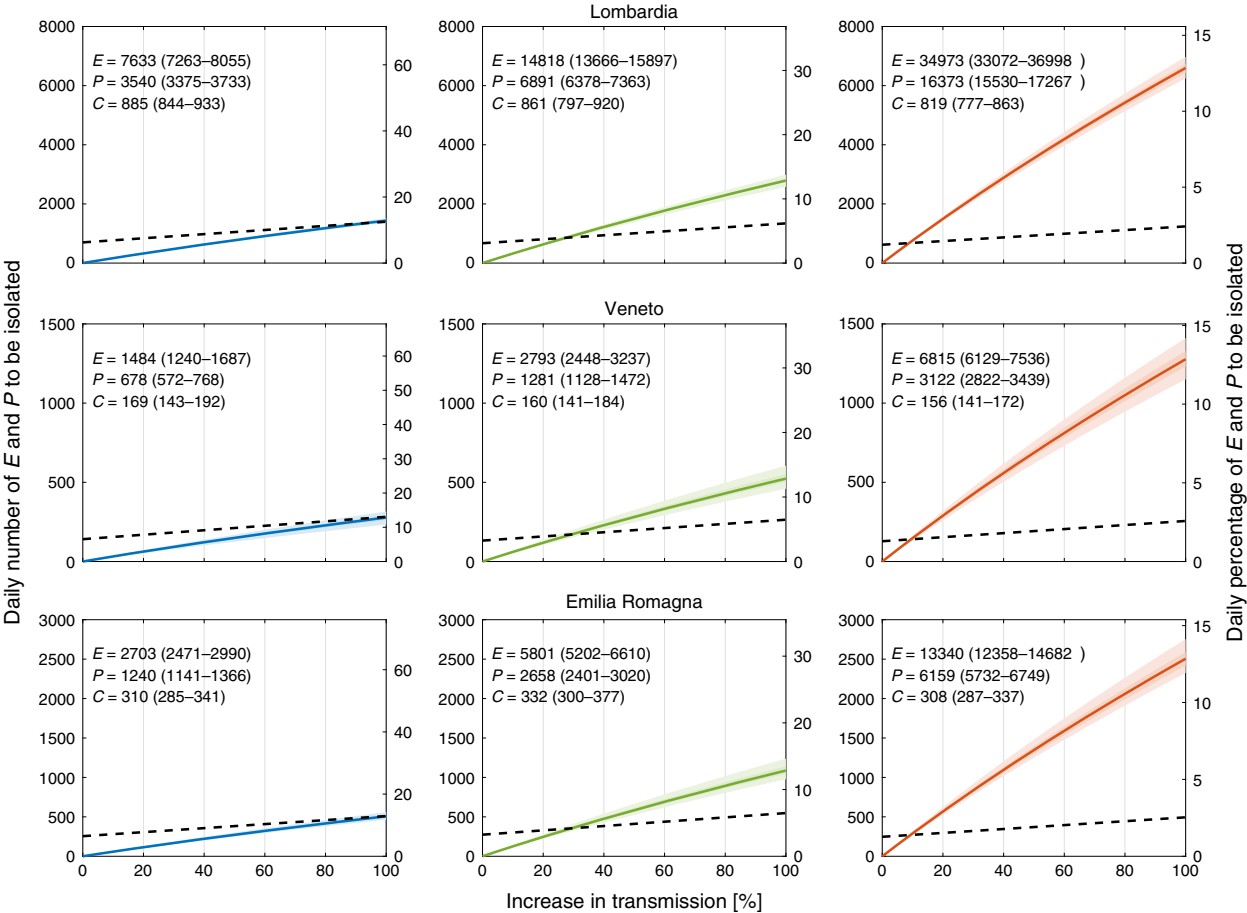

**Fig. 4 Charting the isolation efforts required to control COVID-19 in Italy.** Daily number (left axis) and daily percentage (right axis) of exposed $E$ and individuals at peak infectivity, $P$, to be isolated daily to maintain the epidemic trajectory onto the blue curve in Fig. 2 (corresponding to the baseline scenario) despite the possible increase in transmission induced by the actual release of restrictions (horizontal axes). Different columns refer to different values of the symptomatic fraction $\sigma$: 50% (left, blue), 25% (center, green), and 10% (right, red). Solid lines refer to median values, shaded areas to the 95% (lighter shade), and 50% confidence intervals. In each panel, median and 95% confidence interval of $E$, $P$ and new daily symptomatic cases ($C$) are given (estimates refer to May 4). The dashed black lines indicate the estimated number of $E$ and $P$ individuals that can be isolated by tracing all the infections generated by the new daily symptomatic cases. The other Italian regions are shown in Supplementary Figs. 4, 5, and 6.

The results of our analysis (Fig. 5 and Table 2) show that most regions experienced a decrease of the median transmission rate between the last phase of the lockdown and the following period. Notably, Lombardia, the most severely hit region that accounts for 39% of total Italian cases to date and 54% of those occurred after lockdown relaxation, and Molise are estimated to have experienced an increase in the median transmission (but note that the estimate for Molise, one of the smallest Italian region by number of residents and among the least hit by COVID-19, is marked by large uncertainty). At the country scale, the aggregate count of new daily hospitalized cases is slightly lower than the baseline scenario (Fig. 5).

## Discussion
The results obtained with the data up to the end of lockdown, whose reliability to issue scenarios stems from their capability to closely match the hospitalization data locally and globally, probed in particular the role of inapparent infections by assuming a rather broad range of values of $\sigma$. The highest value ($\sigma = 50\%$) matches the empirical results found by testing for two weeks an entire community (Vo' Euganeo, IT, ~3000 inhabitants)[12], whereas the lowest ($\sigma = 10\%$) is likely to be a lower bound, unachievable in the actual geographic context, because it may reflect also the age structure of a much younger population[38].

Another way to assess the plausibility of the assumed fraction of infections that develop heavy symptoms, is to compare the IFR estimated with different values of $\sigma$. The values of IFR corresponding to $\sigma = 25\%$ and $\sigma = 10\%$ (2% and 0.8%, respectively) bracket the available estimates of IFR for western countries[39]. Thus the median value assumed ($\sigma = 25\%$) seems like a sensible choice to probe the actual role of the unobserved epidemics in Italy.

Social distancing, PPE use, reduced or impeded mobility, and increased awareness led to an overall decrease of transmission of about 65% with respect to the initial uncontrolled epidemic (Fig. 1d). This result is consistent with other estimates obtained using different methods[40,41], and is largely attributable to the implemented measures, and only marginally to acquired immunity. Indeed, even in the most extreme scenario considered here ($\sigma = 10\%$), the acquired immunity would be responsible for less than 15% of the reduction occurred in the most severely hit region (Lombardia, Fig. 3), suggesting that herd immunity is far away even in the hardest hit territories. Seasonality effects[5], not explicitly accounted for here, might also have had a role in the reduction of transmission.

The scenarios shown in Fig. 2 suggest the impact of social distancing, testing, contact tracing and household quarantine on a possible second-wave of the COVID-19 epidemic in Italy

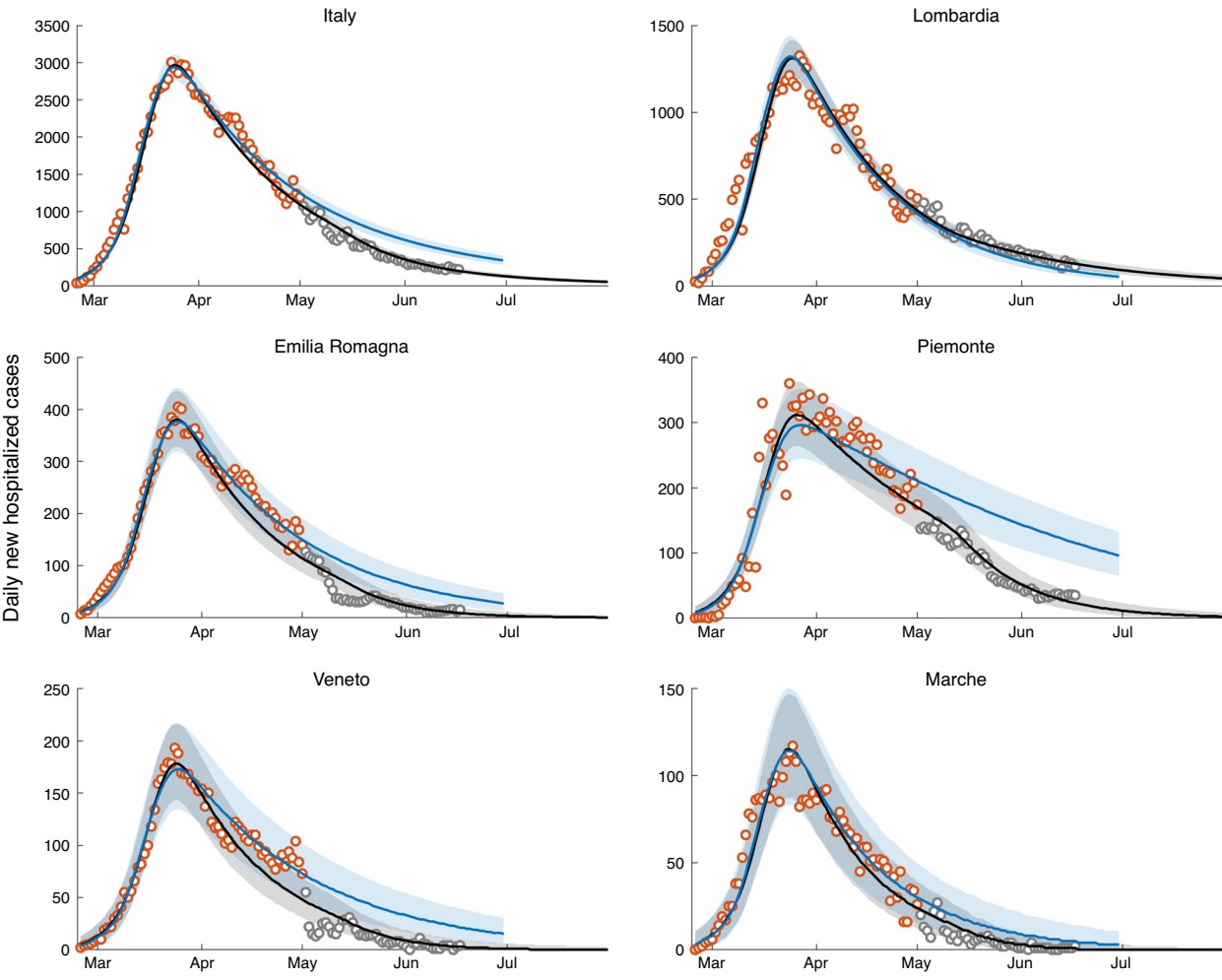

**Fig. 5 Ex-post assessment.** Daily numbers of newly hospitalized cases for Italy and its hardest hit regions. Clockwise from top: Italy, Lombardia, Piemonte, Marche, Veneto, and Emilia Romagna. The remaining regions are shown in Supplementary Fig. 8. Red empty circles represent data available for the projection of the scenarios presented in Fig. 2, gray empty circles show the newly available data. Blue colors (solid line: median results, shaded area: 95% confidence intervals) report here, for the sake of comparison, the baseline scenario already presented in Fig. 2. Black colors show the results of the updated parameter estimation that exploits all data available up to June 17.

**Table 2 Ex-post assessment of transmission rate.**

| Region | $\beta_{P_4}/\beta_{P_3} - 1$ | Region | $\beta_{P_4}/\beta_{P_3} - 1$ |
|---|---|---|---|
| Piemonte | −22% (−26%, −18%) | Marche | −27% (−39%, −15%) |
| Valle D'Aosta | −43% (−69%, −18%) | Lazio | −28% (−34%, −22%) |
| Lombardia | +8% (+6%, +11%) | Abruzzo | −10% (−23%, +2%) |
| Trentino A.A. | −24% (−41%, −8%) | Molise | +8% (−24%, +39%) |
| Veneto | −23% (−31%, −15%) | Campania | −25% (−33%, −18%) |
| Friuli V.G. | −10% (−22%, +2%) | Puglia | −34% (−42%, −26%) |
| Liguria | −17% (−24%, −8%) | Basilicata | −28% (−54%, −1%) |
| Emilia Romagna | −20% (−26%, −14%) | Calabria | −20% (−40%, −2%) |
| Toscana | −17% (−24%, −9%) | Sicilia | −29% (−40%, −21%) |
| Umbria | −36% (−66%, −8%) | Sardegna | −55% (−72%, −36%) |

Changes of the estimated transmission rate after the relaxation of the lockdown for the 20 Italian regions. Median values and 95% confidence intervals.

(see also ref. [36]). The heterogeneous estimates of transmission achieved in the last phase of the lockdown (Fig. 1d) translate in a differential regional response to a possible increase in transmission. Noticeable, the heavily hit region of Lombardia would withstand an increase of +20% of transmission without a rebound of the number of new daily cases (Fig. 2) because our estimates indicate that virus transmission was substantially reduced during the lockdown. By comparison Piemonte, a region that shares with Lombardia a similar susceptibility profile (Fig. 3), might witness a rebound in the same scenario because virus transmission is estimated to still be sustained at the end of the lockdown (Fig. 1d). Such differential spatial effects highlight the relevance of properly accounting for the geography of the disease and to design tailored parameter

estimation frameworks capable of capturing heterogeneity in transmission.

An observed deviation of incoming epidemiological data from the baseline scenario in Fig. 2 (say, towards an unacceptable epidemic trajectory like the purple curve) should raise a red flag and call for control action. Matching the right scenario in real time may be achieved through data assimilation and ensemble Kalman filtering[10]. The continuous update of the estimated state and parameters of the system, in fact, would allow the coupling of feedback and feed-forward controls, thus projecting the number of apparent and inapparent infections at least a latency period ahead of time. Incidentally, we deem this feature a significant advance produced by our method. Indeed, this procedure would provide—in time for action—a reasoned assessment of the actual exposure, in particular the number of exposed and infectious individuals that only models can evaluate. This is tantamount to distinguishing, after lifting the lockdown, between potential and realized transmission. The former is the maximum possible prevention of contagion given a set of rules. The latter is the bulk effect of the effective compliance to precautions associated with the relaxation of the lockdown. Thus, an increase in estimated exposure reflects the actual collective behaviors of mobile individuals, and the collective respect of rules regarding social distancing, PPE adoption, or crowding, to name a few. We argue that realized transmission can only be evaluated from early signs decoded from scenarios implemented through a model akin to ours.

Control may consist in either re-tightening of the containment measures, possibly of the stop-and-go type[5,8], or alternative interventions. Although the strategy adopted during the first phase of the outbreak mostly relied on the isolation and treatment of symptomatic cases, a different mix of interventions is possible and desirable for the second phase. A keystone of such a mix should be an increased isolation effort by tracing[42] and testing[6,34] individuals who have been in close contact with a known infection[26,43], possibly with the help of technological advances like tracing apps[35].

We estimated the isolation target needed to counterbalance an increase in effective transmission (Fig. 4), and to maintain the epidemic trajectory onto the decreasing pattern achieved during lockdown. One way to achieve the required isolation target is to trace the close contacts of daily new symptomatic cases, who are more likely to self-report or be otherwise identified. It should be noted, however, that infected individuals might not immediately test positive (e.g., individuals that are categorized in the exposed compartment $E$ are unlikely to be detected, see, e.g., ref. [44]). Moreover, obtaining test results takes time, therefore this strategy might imply, as a matter of precaution, to isolate, at least temporarily, all close contacts that the symptomatic case has had in the previous days. Tracing and isolating all primary infections generated by the symptomatic cases is a challenging task, because tracing is hardly exhaustive and not all symptomatic cases can be identified. Depending on the extent of the unobserved epidemic, however, isolation of primary infections might not suffice (Fig. 4). Secondary contacts ought to be targeted as well in this case. Testing primary contacts would help identifying actually infected cases, thus refining the tracing of secondary contacts[34]. We also showed that, if the isolation target proves impossible to achieve for the limits of resources and/or logistical reasons, a possible strategy may consist in delaying further relaxations of confinement measures. Our results thus suggest that each Italian region should carefully evaluate its current strategies for tracing, testing and its isolation capacity, to plan and manage the second phase of the epidemic.

The ex-post assessment of the projected scenarios, provided in Fig. 5 and Table 2, shows that the likely increase in contact rate among individuals following the partial relaxation of the restrictive measures that began on May 4 did not lead to a significant increase in transmission rate as of June 17. A notable exception is Lombardia, the most populated Italian region and the one struck the hardest by the epidemic, which is instead estimated to have experienced an increase in the virus transmission. Several intertwined factors could explain these patterns of disease progression after lifting the lockdown. In the following we discuss what we deem most relevant. Restrictive measures have been only partially released: education, from pre-school to higher education, will resume in-presence activities only in September; large gatherings are still forbidden; every commercial activity and workplace is still subject to stringent protocols to ensure social distancing and avoid infections; PPE are mandatory in every indoor setting but one's own household and recommended outdoors whenever preventing social contact is not possible. Altogether, these measures may have contributed significantly to avoid a recrudescence of virus transmission. In particular, evidence accumulates for the crucial role of face masks in reducing transmission[45,46].

Although there is no public data set about contact tracing and isolation, anecdotal evidence and news reports suggest that the effort in such direction has improved in many regional health departments. In our updated estimation (Fig. 5), we did not explicitly account for this effort; however, as detailed in the analysis presented in Fig. 4, increased isolation effort is expected to have the same macroscopic effect as reduced transmission. The abatement of the number of cases achieved during the lockdown period has arguably facilitated the logistics of contact tracing and isolation. It is interesting to note, in fact, that Lombardia—still counting dozens of new daily cases, which in turn makes contact tracing challenging—is the only region showing a significant increase in transmission.

Some authors have suggested that COVID-19 transmission may be subject to seasonal variations in analogy with other human coronaviruses (see, e.g., ref. [5]). However, this hypothesis is still debated among experts, and a clear evidence has not emerged yet. Should warmer, drier weather be proven to actually hinder transmission, such factor could have played a role in Italy during the timeframe analyzed herein.

During lockdown, transmission was reportedly sustained also by epidemic foci in long-term care facilities and hospitals. Succeeding in controlling these mostly local infection chains would have reduced the overall transmission regardless of the restrictive measures. However, also in this case there is not enough granularity in the data to properly support this hypothesis.

Finally, on a more technical side, when virus transmission is slow and occurs in several independent foci, it is possible that infection chains self-extinguish because of demographic stochasticity. Our modeling framework assumes continuous state variables and therefore it does not account for the stochasticity induced by discrete events. As such, our model can overestimate infections when the case count is low.

To keep the epidemic under control, health policy makers should continue to consider a portfolio of interventions that include the re-tightening of confinement measures, possibly with a stop-and-go implementation based also on seasonality effects, or evidence on limits to the acquired immunity[47], and/or the effective isolation of infectious individuals[4,10]. Modeling studies can provide reasoned estimates of the minimum target to be attained. The proper strategy to achieve the isolation target is the domain of public health policy. The complementary use of testing in the control strategy is instead the domain of virology and epidemiology. To both domains, proper modeling scenarios offer information otherwise unavailable.

## Methods

**Epidemiological model.** Here, we use the model $SEPIA$[11]. The model is spatially explicit, i.e., it accounts for the coupled dynamics of a set of $n$ interacting communities. In each community, say $i$ ($i = 1…n$), the model includes the following compartments: susceptible ($S_i$), exposed ($E_i$), peak infectivity ($P_i$), infected with heavy symptoms ($I_i$), asymptomatic/mildly symptomatic ($A_i$), hospitalized ($H_i$), quarantined at home ($Q_i$), recovered ($R_i$), and dead ($D_i$) individuals. The dynamics of transmission is given by:

$$
\begin{aligned}
\dot{S}_i &= -\lambda_i(t)S_i \\
\dot{E}_i &= \lambda_i(t)S_i - \delta_E E_i \\
\dot{P}_i &= \delta_E E_i - \delta_P P_i \\
\dot{I}_i &= \sigma\delta_P P_i - (\eta + \gamma_I + \alpha_I)I_i \\
\dot{A}_i &= (1-\sigma)\delta_P P_i - \gamma_A A_i \\
\dot{H}_i &= (1-\zeta)\eta I_i - (\gamma_H + \alpha_H)H_i \\
\dot{Q}_i &= \zeta\eta I_i - \gamma_Q Q_i \\
\dot{R}_i &= \gamma_I I_i + \gamma_A A_i + \gamma_H H_i + \gamma_Q Q_i \\
\dot{D}_i &= \alpha_I I_i + \alpha_H H_i.
\end{aligned}
\tag{1}
$$

Susceptible individuals ($S_i$) become exposed to the viral agent by contacting individuals who are in any of the three infectious stages, namely peak infectivity, heavily symptomatic or asymptomatic/mildly symptomatic. Frequency-dependent contact rates are assumed, so that exposure is governed by the community-dependent, time-varying force of infection

$$
\lambda_i(t) = \sum_{j=1}^{n} \mathcal{C}_{ij}^S(t) \frac{\sum_{Y\in\{P,I,A\}}\sum_{k=1}^{n}\beta_{Y,k}(t)\mathcal{C}_{kj}^Y(t)Y_k}{\sum_{X\in\{S,E,P,I,A,R\}}\sum_{k=1}^{n}\mathcal{C}_{kj}^X(t)X_k},
$$

where $\mathcal{C}_{ij}^X(t)$ (with $X \in \{S, E, P, I, A, R\}$) is the probability ($\sum_{j=1}^{n}\mathcal{C}_{ij}^X(t) = 1$ for all $i$'s, $X$'s, and $t$'s) that individuals in epidemiological state $X$ who are from community $i$ enter into contact with individuals who are present at community $j$ at time $t$ as either residents or because they are traveling there from community $k$ (note that $i$, $j$, and $k$ may coincide), and $\beta_{Y,j}(t)$ ($Y \in \{P, I, A\}$) are the stage- and time-dependent transmission rates.

Exposed individuals ($E_i$) are latently infected, until they enter the peak infectivity stage (at rate $\delta_E$). This stage has been specifically introduced[11] to account for the clinical and epidemiological evidence indicating that viral shedding peaks just before symptom onset and then declines after the emergence of symptoms or the evolution towards an asymptomatic case[12,15]. Peak infectivity individuals ($P_i$) progress (at rate $\delta_P$) to become (with probability $\sigma$) either symptomatic individuals with heavy clinical symptoms ($I_i$) or (with probability $1 - \sigma$) asymptomatic/mildly symptomatic individuals ($A_i$). Heavily symptomatic infectious individuals exit their compartment if/when (a) they seek treatment at a health-care facility, (at rate $\eta$), following which they may be hospitalized (a fraction $1 - \zeta$) or quarantined at home (a fraction $\zeta$; either ways, they are assumed to be effectively removed from the general community), (b) recover from infection (at rate $\gamma_I$), or (c) die (at rate $\alpha_I$). Asymptomatic/mildly symptomatic individuals ($A_i$) leave their compartment upon recovering from infection (at rate $\gamma_A$). Hospitalized individuals ($H_i$) may either recover from infection (at rate $\gamma_H$) or die (at rate $\alpha_H$). Quarantined (i.e., home-isolated) individuals ($Q_i$) leave their compartment upon recovery (at rate $\gamma_Q$). People recovering from infection or dying because of COVID-19 populate the classes of recovered ($R_i$) and dead ($D_i$) individuals, respectively.

Model (1) is run at the scale of second-level administrative divisions, i.e., provinces and metropolitan cities (107 units as of 2020). Population size in each spatial unit is taken from the official estimates provided yearly (last update: January 1st, 2019) by the Italian National Institute of Statistics (Istituto Nazionale di Statistica, ISTAT; data available at http://dati.istat.it/Index.aspx?QueryId=18460).

The values of the transmission rates ($\beta_{Y,i}(t)$) are dependent on epidemiological status ($Y \in \{P, I, A\}$) as in the original formulation of the model[11]. In addition, they are assumed to be space- and time-dependent to take into account the effects of the various containment measures put in place in the first months of the epidemic (see below for further details).

Spatial coupling is parameterized by using information from the latest nation-wide assessment of mobility fluxes, which was produced by the Italian National Institute of Statistics (ISTAT) in 2011 (data available at https://www.istat.it/it/archivio/139381). For each second-level administrative unit (province), say $i$, two quantities are extracted from the ISTAT data, namely the fraction $p_i$ of mobile people, i.e., the residents of $i$ who defined themselves as commuters, and the fraction $q_{ij}$ of mobile people between $i$ and all other administrative units $j = 1…n$ (including $j = i$). The contact probabilities at the beginning of the epidemic ($t = 0$), $\mathcal{C}_{ij}^X(0)$ ($X \in \{S, E, P, I, A, R\}$) are then defined based on the quantities $p_i$ and $q_{ij}$. Specifically, we assume

$$
\mathcal{C}_{ij}^X(0) = \begin{cases} (1-p_i) + (1-r_X)p_i + r_X p_i q_{ij} & \text{if } i = j \\ r_X p_i q_{ij} & \text{otherwise,} \end{cases}
$$

where the parameter $r_X$ ($0 \leq r_X \leq 1$) describes the fraction of contacts occurring while individuals in epidemiological compartment $X$ are traveling. In other words, for community $i$, the social contacts of non-mobile people (a fraction $1 - p_i$ of the community size), those of mobile people that do not occur during travel (a fraction $1 - r_X$ of total contacts for people in epidemiological compartment $X$) and those associated with mobility for people who travel within their community (a fraction $q_{ii}$ of mobile people) contribute to social mixing within the community. Conversely, the contacts occurring between two different communities, say $i$ and $j$, are a fraction $r_X$ of the total contacts of the individuals in epidemiological compartment $X$, multiplied by the probability $p_i$ that people from $i$ travel (independently of the destination) and the probability $q_{ij}$ that the travel occurs between $i$ and $j$. To account for the effect of the confinement measures, we progressively reduce extra-province mobility according to the estimates obtained through the analysis of data collected through mobile applications[25]. As a conservative assumption, we elaborate near future scenarios maintaining the same level of extra-province mobility estimated during the lockdown, as only few commercial and production activities have resumed and extra-regional mobility is not allowed. For the ex-post assessment instead (Fig. 5), we exploit updated mobility data that became available after the initial submission[48].

**Epidemiological data.** For the calibration of the model, we consider the epidemiological data collected by the Dipartimento della Protezione Civile (data available at https://github.com/pcm-dpc/COVID-19), which are released daily and comprehend: at the regional level, the cumulative numbers of positive, dead and recovered individuals, together with the actual number of positive individuals that were hospitalized or under quarantine at home; at the province level, the cumulative numbers of positive cases.

Due to the strong space-time variations in the number of tests performed, the most trustworthy variable to monitor the outbreak is the daily number of hospitalizations, in the following indicated with $H^{in}$. This quantity corresponds to the flux $(1 - \zeta)\eta I$ in the $H$ compartment of our model, and grants a straightforward link between data and model variables. However, $H^{in}$ is not directly provided in the online data, and we thus adopt a stochastic approach to derive $H^{in}$ combining data regarding the number of hospitalized individuals and deaths, and estimating distribution of delays between hospitalization and death or discharge.

At any given day $k$, $H_k^{in}$ is obtained by the observed variations in the daily number of hospitalized individuals, $H_k - H_{k-1}$, plus the daily deaths $D_k^{out} = D_k - D_{k-1}$, and the number of individuals discharged from the hospital, here indicated with $R_k^{out}$. Under the assumption that the recorded deaths for COVID-19 are all from the hospitals, $R_k^{out}$ is obtained by modeling as random variables the days $\tau$ spent in a hospital before death, whose probability density function (PDF) is indicated with $p_D(\tau)$, and the time in a hospital before discharge, whose PDF is indicated with $p_R(\tau)$.

Our procedure consists of the following steps. Sampling a random value from $p_D(\tau)$ for each individual in $D^{out}$, we obtain the days of entrance of individuals that will die, thus the sequence $D_k^{in}$. We estimate the number of individuals entering on day $k$ that will be eventually discharged as:

$$
R_k^{in} = H_k^{in} - D_k^{in}.
$$

Then $R_k^{out}$ is obtained by sampling an exit time from $p_R(\tau)$ for each individual in $R_k^{in}$. Finally:

$$
H_k^{in} = H_k - H_{k-1} + D_k^{out} + R_k^{out}.
$$

Reports by ISS indicate that for COVID-19 casualties the median residence time at a hospital is about 8 days for patients that accessed ICU, and 5 days without ICU[49]. We use this information to parameterize the distribution $p_D(\tau)$ as a gamma distribution of mean 7 and coefficient of variation 0.5 (hence, a median of 6.42 days; 0.05–0.95 quantiles: 2.39–13.56 days). We also assume that $p_R(\tau)$ follows a gamma distribution of mean 14 and coefficient of variation 0.5, which has a median of 13.7 days (0.05–0.95 quantiles: 4.78–27.14 days), in agreement with the recovery rate previously estimated[11].

Final data adopted for parameter estimation of the model is the median over 100 random generations of $H_k^{in}$, downscaled to the province level and smoothed by using a moving average of 7 days. A sensitivity analysis of $H_k^{in}$ on the parameters of the $p_D(\tau)$ and $p_R(\tau)$ showed that the time series considered have only marginal variations.

**Parameter estimation.** The effect of the containment measures was parameterized by assuming that the transmission parameters ($\beta_P$, $\beta_I$ and $\beta_A$) had a sharp decrease after the containment measures announced on February 24 and March 8[11]. We update here such description to fully account for the set of progressively more restrictive measures that were introduced from March 8 to March 22, when also non-essential industrial and production activities were stopped. We describe the temporal changes in the $\beta_P$'s (the remaining transmission parameters, $\beta_I$ and $\beta_A$, are assumed to be proportional to $\beta_P$, see Table 1) using 4 values: The value before February 24 ($\beta_{P_0}$), the values achieved right after (within two days) the measures introduced on February 24 ($\beta_{P_1}$) and the first set of lockdown measures implemented on March 8 ($\beta_{P_2}$). Finally, we assume that due to the progressive implementation of the lockdown and the introduction of more restrictive measures, the

transmission rates further linearly decreased from March 10 to March 22, eventually achieving the value ($\beta_{P_3}$), which is then held constant. We let $\beta_{P_3}$ vary among different Italian regions to reflect possible heterogeneity in disease transmission. Specifically, we estimate the hyperparameters controlling the prior of the parameters $\beta_{P_3}/\beta_{P_2}$ (a Gaussian distribution truncated between 0 and 1) in a hierarchical Bayesian framework. In the ex-post assessment of our scenarios, we introduced an additional parameter, $\beta_{P_4}$, that quantifies the transmission after the relaxation of the restrictive measures (i.e., after May 4). We let also $\beta_{P_4}$ to possibly vary among the different Italian regions.

We impose an initial condition of one exposed individual in the province of Lodi (where the first cases were reported) $\Delta t_0$ days before February 24. We further estimate also the initial condition of the exposed compartment in each province to account for the possible seeding effect occurred during this period[11].

Parameters are estimated by comparing data and simulation of the flux of hospital admissions (($(1 - \zeta)\eta I$) at the provincial scale. We assume that each data point follows a negative binomial distribution with mean $\mu$, equal to the value predicted by the model, and variance equal to $\omega\mu$ (NB1 parametrization[50,51]). Parameter values are summarized in Table 1.

Model (1) assumes that recovered individuals are immune and that immunity loss is negligible within the simulation horizon considered herein (160 days). We performed additional simulations assuming a fast waning immunity (average duration equal to three months, as suggested by recent evidence[47]). Simulations up to July 31 are almost indistinguishable from those that assume permanent immunity (Supplementary Fig. 9).

**The effect of testing and isolation**. Following lockdown release, the expected increase in the transmission rates can be compensated by isolation of cases. Clinical and epidemiological evidence suggests that viral shedding peaks at the end of the latent period, and that shedding rapidly declines after the symptoms' onset or the evolution towards an asymptomatic case[15]. Moreover, viral shedding is similar regardless of the emergence of symptoms in the disease course of a patient[12]. This evidence suggests that isolation should be more effective if targeted at individuals in the exposed, $E$, and peak infectivity, $P$, compartments of the model. Therefore, we focus on these individuals as priority targets for isolation.

When isolation is enforced, two out-fluxes, $\rho_{E,i}E_i$ and $\rho_{P,i}P_i$, must be considered from the exposed and peak infectivity compartments, respectively. The parameters $\rho_{E,i}$ and $\rho_{P,i}$ (d$^{-1}$) represent the community-dependent rate at which infected individuals in the $E_i$ and $P_i$ classes are effectively isolated from the community. For the sake of simplicity, we assume $\rho_E = \rho_P = \rho$. Also, individuals isolated are simply removed from the community, without any further consideration of their clinical trajectories, which is deemed reasonable considering the relatively short timespan of the simulations performed here.

We estimate for each province the percentage (i.e., $\rho_i$) and the corresponding number of individuals ($\rho_i(E_i + P_i)$) that should be isolated daily to counterbalance the increase in transmission due to the loosening of containment measures, so as to maintain the same level of trasmissivity achieved during the lockdown. In analogy to the analyses presented above, we repeat the estimation of the isolation effort under the three different assumptions about the heavy symptomatic fraction: $\sigma = 25\%$, $50\%$, and $10\%$.

**Reporting summary**. Further information on research design is available in the Nature Research Reporting Summary linked to this article.

## Data availability

All data used in this manuscript are publicly available. COVID-19 epidemiological data for Italy are available at https://github.com/pcm-dpc/COVID-19. Mobility data at municipality scale are available at https://www.istat.it/it/archivio/139381. Population census data are available at http://dati.istat.it/Index.aspx?QueryId=18460. Processed data that are used in the analysis are available in the public repository https://github.com/COVID-19-routes/geography-paper.

## Code availability

The analysis of the data was performed via a custom code developed in Matlab programming language (version R2018a) and available at the repository: https://github.com/COVID-19-routes/geography-paper.

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

## Acknowledgements

E.B., D.P., and A.R. acknowledge funding from Fondazione Cassa di Risparmio di Padova e Rovigo (IT) through its Grant 55722 (April 2020). E.B. acknowledges the support provided by the European Horizon2020 project Nunataryuk (773421).

## Author contributions

E.B., M.G., and A.R. were responsible for conceiving the work. E.B., D.P., and L.M. were responsible for numerical simulations and model parameter estimation. E.B., L.M., D.P., S.M., R.C., M.G., and A.R. were responsible for data analysis and statistics and writing the manuscript.

## Competing interests

The authors declare no competing interests.
