## [Peer Review File · Nature Communications]

REVIEWER COMMENTS

Reviewer #1 (Remarks to the Author):

- The paper is very interesting: it presents a detailed model of the evolution of COVID that takes care of the quite diverse spread of the illness in different regions of Italy. This is a crucial feature that has to be addressed.

The paper also contains a careful study of the possible effects of lifting the restriction.

- The results are novel: they are relevant in order to construct realistic models of epidemics, where geography takes a very important role.

- As far as I know, the results are original and the reference list is exhaustive.

- I consider the work convincing.

- I hope that the paper will influence thinking in the fields. I have seen too many papers that arrive at wrong conclusions because they have neglect geographic effects.

Overall I suggest that the paper should be published

Reviewer #2 (Remarks to the Author):

This is an excellent paper that extends the work described in the earlier paper by Gatto et al that developed a spatial model for Covid-19 in Italy. here the model is used to examine the response of the epidemic to the relaxation of lockdown and a return to work. The paper does an exemplary job of illustrating the sensitivity of a second wave of infection to different forms of social relaxation.

I highly recommend that it be published - it is an elegant and well parameterized model that provides vital public health insights of direct relevance to Italy as well as to other nations and regions.

A few changes might be instructive:

1) if possible update model to at least June 1st and extrapolate forward from a date as close to publication as possible.

2) The model assumes that recovered individuals remain resistant, could this assumption be relaxed to see what happens if immunity only lasts a matter of months?

3) People become infected by exposure to hosts in different stages of infection, is it possible to include a diagram showing the relative contribution of different classes of infected hosts to new infections through time? This would help strengthen the argument about the efficacy and importance of maintaining different types of lockdown and social distancing.

Reviewer #3 (Remarks to the Author):

The paper reuses the SEPIA spatially-explicit epidemiological model, already proposed in the previous work [9] by the authors, to simulate the effect of lockdown weakening on the Italian COVID-19 epidemic.

The baseline trajectory with lockdown kept in place is estimated and compared with the possible evolutions in presence of relaxation of the lockdown, considering situations where the lockdown release makes the transmission rate increase of 20% and 40%.

A comparative analysis in the different Italian regions is conducted.

Also the importance of isolation is highlighted, discussing the fraction of exposed and infectious individuals that should be isolated to compensate for the increased transmission rate.

The analysis looks correct and very solid.

The novelty is somehow limited, because the model has been already proposed and here is just used to assess post-lockdown (instead of lockdown) scenarios, while scenarios with weakened lockdown and strategies for safe lockdown relaxation have been already proposed in recent literature (cf. [5], [6], but also <https://arxiv.org/abs/2003.09930>).

However, this work can be timely and interesting if it takes advantage of all the new data available up to today to assess the actual outcome of the initial region-wide lockdown weakening.

- The lockdown has been released region-wise in the past weeks. Can the authors compare the effect of this release with the scenarios they outline?

I believe it is mandatory to include the most recent data and compare them with the model predictions.

In particular, it would be important to quantify the increase in the transmission rate that actually occurred so far, based on the newly available epidemic data.

- Looking further to the future, what does the model predict for a scenario where regional and national borders are reopened?

- The scenarios with a weakened lockdown look quite different in regions with an apparently similar situation. For instance, in Figure 2, it seems that the epidemic in Lombardia would explode way less than in Piemonte with a 40% increase in the transmission rate, even though the remaining fraction of susceptibles is comparable in the two regions according to Figure 3.

How can this be explained?

Also, what justifies the apparently milder case explosion in Lombardia according to Figure 2, while that region still accounts for the vast majority of Italian cases also after the lockdown has been weakened (regionally) for some weeks?

It would be nice to see a discussion/analysis of the differences between regions, since the spatial description is presented as a strong plus of the SEPIA model.

- Concerning the effect of increased testing / tracing combined with lockdown release, it would be interesting to compare the results obtained using the spatial SEPIA model with the scenarios already outlined in [6]. Would there be significant regional differences, which the spatial model can reveal? If so, how can these differences be explained?

- As for the feasibility of the proposed strategy, it is suggested to isolate in particular individuals in the exposed and infection-peak compartments. Can exposed individuals, who are at a very early stage of the infection, be detected with a nasal swab? Maybe we run the risk of false negatives because at such an early stage the viral load is too small to be detected.

Discussing this aspect would be important.

- In the final part of the discussion, the authors hint to stop-and-go confinement measures. What has their model to say about this possible scenario?

RESPONSE TO REVIEWERS

Detailed answers to the Referees' comments are reported below. *Italic* fonts refer to Reviewers' statements, our comments follow in Roman type. We include as a separate file also the manuscript with all tracked changes.

Reviewer #1

- *The paper is very interesting: it presents a detailed model of the evolution of COVID that takes care of the quite diverse spread of the illness in different regions of Italy. This is a crucial feature that has to be addressed. The paper also contains a careful study of the possible effects of lifting the restriction.*
- *The results are novel: they are relevant in order to construct realistic models of epidemics, where geography takes a very important role.*
- *As far as I know, the results are original and the reference list is exhaustive.*
- *I consider the work convincing.*
- *I hope that the paper will influence thinking in the fields. I have seen too many papers that arrive at wrong conclusions because they have neglect geographic effects.*

Overall I suggest that the paper should be published.

We would like to thank this Reviewer for their positive assessment of our manuscript.

Reviewer #2

This is an excellent paper that extends the work described in the earlier paper by Gatto et al that developed a spatial model for Covid-19 in Italy. Here the model is used to examine the response of the epidemic to the relaxation of lockdown and a return to work. The paper does an exemplary job of illustrating the sensitivity of a second wave of infection to different forms of social relaxation.

I highly recommend that it be published - it is an elegant and well parameterized model that provides vital public health insights of direct relevance to Italy as well as to other nations and regions.

The authors are pleased to see that the manuscript was well received and assessed also by the second Reviewer.

A few changes might be instructive: 1) if possible update model to at least June 1st and extrapolate forward from a date as close to publication as possible.

We welcomed this suggestion, and revised the paper including the new epidemiological data (up to June 17) that became available after the initial submission. Specifically, in the new Figure 5 we compare the actual evolution of the outbreak with the three scenarios provided in Figure 2. Data are consistent with the baseline scenario. Interestingly, the likely increase in contact rate after partially lifting the lockdown in Italy has not resulted in a significant increase in transmission so far. We then re-estimated the original model parameters incorporating also the new data available and calibrated an additional parameter controlling the change in transmission after lifting of the lockdown (Figure 5 and Table 2 of the main text). The model simulations are then used to project scenarios up to the end of July 2020. The last part of the paper is now devoted to discuss these findings.

2) The model assumes that recovered individuals remain resistant, could this assumption be relaxed to see what happens if immunity only lasts a matter of months?

The problem of immunity loss is very important and should be tackled with care. On the one hand, in fact, SARS-CoV-2 is subject to quite rapid evolution at the origin of a wide genetic diversity [1]. This aspect could be interpreted as favorable to the loss of immunity by the recovered hosts (see for example evidence of short-lasting immunity documented in [2]). On the other hand, the possible cross-reactivity responses shown by individuals who have been exposed to similar (yet not identical) viral strains may originate complex dynamics (as in the transmission of influenza, for example) and need to be further investigated for coronaviruses (see [3]).

As for the model we analyzed here, for all the three assumptions regarding the fraction σ of infected individuals showing heavy symptoms, the percentage of the population who is still susceptible remains quite large, as shown in Figure 3 of the main text. This suggests that a possibly short-lived immunity would have played a minor role in the dynamics of the outbreak so far. To better corroborate this speculative conclusion and provide the Reviewer with quantitative answers on this specific point, we have re-run simulations with a model in which we included loss of acquired immunity in the simplest possible way, i.e. assuming that all recovered individuals become fully susceptible to the virus again after an average timespan $1/\nu$, and without considering

any cross-reactive immune responses. The updated model equations would then read as:

$$\begin{aligned}
\dot{S}_i &= \mu(N_i - S) - \lambda_i(t)S_i + \nu R_i \\
\dot{E}_i &= \lambda_i(t)S_i - (\delta_E + \mu)E_i \\
\dot{P}_i &= \delta_E E_i - (\delta_P + \mu)P_i \\
\dot{I}_i &= \sigma \delta_P P_i - (\eta + \gamma_I + \alpha_I + \mu)I_i \\
\dot{A}_i &= (1 - \sigma)\delta_P P_i - (\gamma_A + \mu)A_i \\
\dot{H}_i &= (1 - \zeta)\eta I_i - (\gamma_H + \alpha_H + \mu)H_i \\
\dot{Q}_i &= \zeta \eta I_i - (\gamma_Q + \mu)Q_i \\
\dot{R}_i &= \gamma_I I_i + \gamma_A A_i + \gamma_H H_i + \gamma_Q Q_i - (\nu + \mu)R_i \\
\dot{D}_i &= \alpha_I I_i + \alpha_H H_i ;
\end{aligned} \tag{1}$$

where ν represents the immunity loss rate. For completeness, in equations 1 we included also the processes of natality and mortality (for causes other than COVID-19) as they also contribute to the replenishment of the susceptible pool. Specifically, we include a mortality rate μ and a constant recruitment μN_i , where N_i is the population size of community i . The rate μ is here set for each community as the inverse of the average lifetime in Italy (83 years). Figure R1 shows model results obtained by assuming permanent immunity (median result, blue color) and a 3-month long immunity (median and 95% confidence interval, reddish colors) through sampling the posterior distribution used in Figure 5 of the main text. It can be noticed that the two simulations up to July 2020 are almost indistinguishable. We have added a note in the Methods section and a Supplementary Figure to report this result.

We further explore a scenario in which transmission would gradually climb back to the level observed at the beginning of the lockdown (value β_{P_2}) by September 1, leading to a second wave and the need for another lockdown starting on October 1 (gray shaded area, the same scenario is used to simulate stop-and-go lockdown in response to points raised by Reviewer #3 below). In such scenario, a short-lived immunity would indeed play a role in absence of cross-reactivity from already recovered hosts, because the pool of susceptible individuals would be almost completely replenished at the beginning of the second wave, leading to an outbreak larger than the first one. However, since the current manuscript is not focused on long-term dynamics, and the epidemiological projections obtained in this simulation scenario rely on speculative assumptions, we decided not to include these (nontrivial, and potentially interesting) results in the revised manuscript.

3) People become infected by exposure to hosts in different stages of infection, is it possible to include a diagram showing the relative contribution of different classes of infected hosts to new infections through time? This would help strengthen the argument about the efficacy and importance of maintaining different types of lockdown and social distancing.

Figure R2 below reports the useful information suggested by this Reviewer. According to the estimated parameters, 86% of the infections occur through contact with hosts during their peak of infectivity stage (P), while 12% and 2% are transmitted after the host has transited to the A and I states, respectively. During the initial phase of the outbreak, the transmission is even more dominated by peak-infectivity hosts. We have included a comment about these results in the revised version of the manuscript where we highlight the importance of isolating infected people early on in the course of disease, i.e. while in stages E or P (lines **156-158**)

Overall, we wish to thank Reviewer #2 for their favourable assessment of our manuscript and insightful comments which we believe resulted in a significantly improved manuscript.

Figure R1: Effect of immunity loss.

Figure R2: Apportionment of infections according to their causative infectious class.

Reviewer #3

The paper reuses the SEPIA spatially-explicit epidemiological model, already proposed in the previous work [9] by the authors, to simulate the effect of lockdown weakening on the Italian COVID-19 epidemic. The baseline trajectory with lockdown kept in place is estimated and compared with the possible evolutions in presence of relaxation of the lockdown, considering situations where the lockdown release makes the transmission rate increase of 20% and 40%. A comparative analysis in the different Italian regions is conducted. Also the importance of isolation is highlighted, discussing the fraction of exposed and infectious individuals that should be isolated to compensate for the increased transmission rate. The analysis looks correct and very solid. The novelty is somehow limited, because the model has been already proposed and here is just used to assess post-lockdown (instead of lockdown) scenarios, while scenarios with weakened lockdown and strategies for safe lockdown relaxation have been already proposed in recent literature (cf. [5], [6], but also <https://arxiv.org/abs/2003.09930>). However, this work can be timely and interesting if it takes advantage of all the new data available up to today to assess the actual outcome of the initial region-wide lockdown weakening.

The authors are pleased to read that also this third Reviewer found our analysis “*correct and very solid*”. May we also note that this manuscript does not simply reuse the spatially explicit model (Ref [9] of the submitted paper) to account for new data over a longer temporal scale. Rather, the spatially-explicit approach is here extended with new tools, such as improved Bayesian estimation to deal with heterogeneous regional transmission and, key to the goal of the present paper, a novel focus on the estimation of isolation targets, i.e. a quantification of how many (mostly inapparent) cases would need to be identified in order to keep disease transmission under control. Following also a somehow similar suggestion by Reviewer #2, the revised version of the manuscript provides an ex-post assessment by comparing newly available data with the three scenarios presented in Figure 2. Moreover, we re-estimated the model parameters to assess the change in transmissivity after the lifting of lockdown, and discussed the results.

We agree with Reviewer #3 that safe lockdown relaxations have been already proposed in the literature. References [5] and [6] cited in the original manuscript are noteworthy for different reasons. Kissler et al. explore scenarios assuming seasonality and acquired immunity, but do not attempt to closely match observed epidemiological patterns. On the other hand, the spatially-implicit analysis proposed by Giordano et al. was performed during the initial phase of the disease and includes scenarios of lockdown release after April 5. Although very relevant and noteworthy when the paper by Giordano and coauthors was published, those scenarios appear to be no longer realistic at the time of our writing (lockdown has been released on May 4). However, all the previous papers are acknowledged as important forerunners from the methodological viewpoint. We thank this Reviewer for pointing us to the work by Bin et al.

(<https://arxiv.org/abs/2003.09930>), which we were not aware of, and which has been included in the reference list of the revised manuscript.

- The lockdown has been released region-wise in the past weeks. Can the authors compare the effect of this release with the scenarios they outline? I believe it is mandatory to include the most recent data and compare them with the model predictions. In particular, it would be important to quantify the increase in the transmission rate that actually occurred so far, based on the newly available epidemic data.

We agree with Reviewer #3 and, as already mentioned in the previous response, we have included the newly available data and provided an ex-post assessment of our scenarios in the revised version of the manuscript (see the new Figure 5). Moreover, we re-estimated parameters to quantify the change in transmission occurred after lockdown release (Table 2 of the revised version of the manuscript). We estimated an increase in transmission of around 8% in Lombardy (the most severely hit region that accounts for the 39% of the total number of cases and for the 54% of the cases after partially releasing the lockdown). Other regions show instead no significant changes or a decrease in transmission. We now devote the last part of the manuscript to discuss these findings.

- Looking further to the future, what does the model predict for a scenario where regional and national borders are reopened?

Regional borders were reopened on June 3, thus the data used in the revised version (up to June 17) already include a two-weeks phase of increased regional mobility. The issue of reopening transnational mobility after most countries controlled the outbreak through restrictive measures is indeed a very interesting, yet completely new problem; and one that requires a very different approach (global or at least pan-European scale). As stated upfront in the originally submitted material, no seeding of disease from outside the national borders was considered. Not only does this seem quite reasonable an assumption for the specific time horizon considered in the manuscript, but also it would be rather difficult and subject to large uncertainty to account for possible new flare-ups caused by mobility from international pandemic foci.

- The scenarios with a weakened lockdown look quite different in regions with an apparently similar situation. For instance, in Figure 2, it seems that the epidemic in Lombardia would explode way less than in Piemonte with a 40% increase in the transmission rate, even though the remaining fraction of susceptibles is comparable in the two regions according to Figure 3. How can this be explained? Also, what justifies the apparently milder case explosion in Lombardia according to Figure 2, while that region still accounts for the vast majority of Italian cases also after the lockdown has been weakened (regionally) for some weeks? It would be nice to see a discussion/analysis of the differences between regions, since the spatial description is presented as a strong plus of the SEPIA model.

The differences among the responses of the different regions to the same scenarios of increased transmission are explained by the heterogeneity of the conditions achieved during the last phase of the lockdown (transmission parameter β_{P_3}). We have indeed specifically introduced a hierarchical Bayesian approach to allow for this possible heterogeneity and better grasp the emerging differences in the transmission occurring during the lockdown in the various Italian regions. As shown in Figure 1d of the originally submitted manuscript, Lombardia was one of the regions that achieved the lower transmission rate during the last phase of the lockdown. Therefore, a 40% increase in transmission would have had a lesser impact there than, say, in Piemonte.

We agree with this Reviewer that this particular asset of the proposed model, i.e. the handling of spatial processes and heterogeneity, was not sufficiently described and highlighted in the submitted manuscript and we have therefore expanded the revised text to properly emphasize it and discuss the ensuing results (lines 249-259). The authors wish to thank this Reviewer for pointing this out.

- Concerning the effect of increased testing / tracing combined with lockdown release, it would be interesting to compare the results obtained using the spatial SEPIA model with the scenarios already outlined in [6]. Would there be significant regional differences, which the spatial model can reveal? If so, how can these differences be explained?

Giordano et al. already showed that an increased diagnostic (and thus isolation) effort can curb the epidemic curve. We have now cited this paper also in this context, and we thank this Reviewer for pointing this out. However, Giordano et al. did not report on the tradeoff between transmission and isolation effort as we did in Figure 4 of the manuscript, and thus a direct comparison is unfeasible. Moreover, the exercise performed by Giordano and coauthors refers to the first phase of the outbreak while ours focuses on the tail. The numbers reported in Figure 4 refer to the isolation effort required to offset an increase of transmission with respect to the one achieved during lockdown. Owing to this choice (which we deemed and still deem the easiest interpretable message), much of the heterogeneity among regions (as illustrated in Figure 1d) is ironed out. Should we show, say, the effort required to prevent a resurgence of cases, the spatial heterogeneity of the effort among regions would be much more evident. As explained in the response to the previous point, this heterogeneity is due to the different levels of virus transmission achieved by the different regions in the last phase of the lockdown as estimated by our Bayesian framework.

- As for the feasibility of the proposed strategy, it is suggested to isolate in particular individuals in the exposed and infection-peak compartments. Can exposed individuals, who are at a very early stage of the infection, be detected with a nasal swab? Maybe we run the risk of false negatives because at such an early stage the viral load is too small to be detected. Discussing this aspect would be important.

This Reviewer raises an important point. Exposed individuals are likely difficult to detect by nasal swab test (see e.g. [4]). In the originally submitted version of the manuscript, we reported on this issue at this point: *“However, as infected individuals might not immediately test positive, and because obtaining test results takes time, this strategy might imply as a matter of precaution to isolate, at least temporarily, all close contacts that the symptomatic case has had in the previous days”*. Indeed, we stressed that the strategy focuses on the isolation of close contacts of symptomatic infected. Testing of such contacts is seen as a complementary tool that may be useful to allow releasing in advance the mandate to self-isolate or, more importantly, to identify actual infections and search for secondary contacts, as highlighted few lines below: *“Testing primary contacts would help identifying actually infected cases, thus refining the tracing of secondary contacts”*. In general, to reach the isolation target estimated in Figure 4, more individuals need to be isolated. For this reason, we believe that comparing such target with the one that can be achieved by isolating all close contacts of the symptomatic cases, as done in Figure 4, is particularly useful and provides a more operational results, rather than a pure modelling exercise. At any rate, we agree that this is a very important point and we have emphasized it more in the revised version of the manuscript (lines 290-300)

- In the final part of the discussion, the authors hint to stop-and-go confinement measures. What has their model to say about this possible scenario?

Our hint at the end of the paper was to indicate the flexibility of the model, not to anticipate results that would require another dedicated manuscript. The proposed model can in fact readily be used to simulate stop-and-go confinement measures. An example is reported in Figure R3 where we make the hypothesis that the transmission rate will gradually (i.e. linearly) climb back to the value estimated at the beginning of the lockdown (i.e. β_{P_2}) by say September 1, leading then to a second wave of infections. From that date on, we assumed a 30-days period of *go* (β_{P_2} transmission) alternated with a 30 days of *stop* (β_{P_4} transmission). If a policy of that kind were implemented, the epidemic would develop with repeated peaks until the acquisition of herd immunity (see e.g. the trajectory of Lombardia where the susceptible fraction is estimated to be the lowest). However, different choices for the stop-and-go phases and/or different baseline transmission levels could lead to either a more marked decline or outbreaks of growing magnitude, as illustrated also by the manuscript pointed out above by this Reviewer [5]. It would be straightforward also to simulate scenarios implementing stop-and-go confinement measures with a heterogeneous geographic deployment, impose selective mobility restrictions, or introduce *stops* based on the current state of the population. Indeed, many scenarios should be investigated and compared for a compelling analysis. However, such an interesting study would really depart from the main goal of the manuscript; and we have therefore decided to not include this further analysis in the current paper.

In conclusion, we wish to thank Reviewer #3 for their stimulating remarks.

Figure R3: Simulation example of a stop-and-go lockdown strategy. Symbols as in Figure 2 and 5 of the main text. Gray shaded areas indicate periods where restrictive measures are applied.

References

1. Phan, T. Genetic diversity and evolution of SARS-CoV-2. *Infection, genetics and evolution* **81**, 104260 (2020).
2. Edridge, A. W. *et al.* Coronavirus protective immunity is short-lasting. *medRxiv*. doi:10.1101/2020.05.11.20086439. eprint: <https://www.medrxiv.org/content/early/2020/06/16/2020.05.11.20086439.full.pdf>. <https://www.medrxiv.org/content/early/2020/06/16/2020.05.11.20086439> (2020).
3. Huang, A. T. *et al.* A systematic review of antibody mediated immunity to coronaviruses: antibody kinetics, correlates of protection, and association of antibody responses with severity of disease. *medRxiv*. doi:10.1101/2020.04.14.20065771. eprint: <https://www.medrxiv.org/content/early/2020/04/17/2020.04.14.20065771.full.pdf>. <https://www.medrxiv.org/content/early/2020/04/17/2020.04.14.20065771> (2020).
4. Kucirka, L., Lauer, S., Laeyendecker, O., Boon, D. & Lessler, J. Variation in False Negative Rate of RT-PCR Based SARS-CoV-2 Tests by Time Since Exposure. doi:10.1101/2020.04.07.20051474. medRxiv: 2020.04.07.20051474 (Apr. 10, 2020).
5. Bin, M. *et al.* *On fast multi-shot COVID-19 interventions for post lock-down mitigation* <https://arxiv.org/pdf/2003.09930.pdf> (2020).

REVIEWERS' COMMENTS:

Reviewer #2 (Remarks to the Author):

This is an excellent paper and the authors have responded in a totally satisfactory way to all my comments.

There is increasing evidence that immunological resistance does not last very long, so it would be good to consider this, but this is a task for a later paper and I have no doubt the authors may already be working on this.

This should be published as quickly as possible, it has important implications for what happens in the second half of this year.

Reviewer #3 (Remarks to the Author):

All my comments have been adequately addressed and I believe the paper has been significantly improved by including the most recent data, better highlighting the spatial insight that the model provides, and broadening the discussion.

I highly recommend acceptance.

POINT BY POINT RESPONSE TO REVIEWER COMMENTS

REVIEWER 2: This is an excellent paper and the authors have responded in a totally satisfactory way to all my comments.

There is increasing evidence that immunological resistance does not last very long, so it would be good to consider this, but this is a task for a later paper and I have no doubt the authors may already be working on this.

This should be published as quickly as possible, it has important implications for what happens in the second half of this year.

The authors wish to thank this Reviewer for their favourable comments.

Regarding the duration of the immunological resistance, we have now included the latest evidence when discussing the figure where the effect of the such duration is assessed (Methods and Supplementary Figure 9).

REVIEWER 3: All my comments have been adequately addressed and I believe the paper has been significantly improved by including the most recent data, better highlighting the spatial insight that the model provides, and broadening the discussion.

I highly recommend acceptance.

The authors wish to thank this Reviewer for their favourable assessment.